# Viruses in saliva from sanctuary chimpanzees (*Pan troglodytes*) in Republic of Congo and Uganda

Emily Dunay[1], Joshua Rukundo[2], Rebeca Atencia[3], Megan F. Cole[4], Averill Cantwell[5], Melissa Emery Thompson[4], Alexandra G. Rosati[5,6], Tony L. Goldberg[1]*

1 Department of Pathobiological Sciences, School of Veterinary Medicine, University of Wisconsin-Madison, Madison, Wisconsin, United States of America, 2 Ngamba Island Chimpanzee Sanctuary / Chimpanzee Trust, Entebbe, Uganda, 3 Jane Goodall Institute Congo, Pointe-Noire, Republic of Congo, 4 Department of Anthropology, University of New Mexico, Albuquerque, New Mexico, United States of America, 5 Department of Psychology, University of Michigan, Ann Arbor, Michigan, United States of America, 6 Department of Anthropology, University of Michigan, Ann Arbor, Michigan, United States of America

* tony.goldberg@wisc.edu

**Data Availability Statement:** Viral nucleotide sequences are available in GenBank under accession numbers OP934203-OP934225. Metadata is available under BioProject

## Abstract

Pathogen surveillance for great ape health monitoring has typically been performed on non-invasive samples, primarily feces, in wild apes and blood in sanctuary-housed apes. However, many important primate pathogens, including known zoonoses, are shed in saliva and transmitted via oral fluids. Using metagenomic methods, we identified viruses in saliva samples from 46 wild-born, sanctuary-housed chimpanzees at two African sanctuaries in Republic of Congo and Uganda. In total, we identified 20 viruses. All but one, an unclassified CRESS DNA virus, are classified in five families: *Circoviridae*, *Herpesviridae*, *Papillomaviridae*, *Picobirnaviridae*, and *Retroviridae*. Overall, viral prevalence ranged from 4.2% to 87.5%. Many of these viruses are ubiquitous in primates and known to replicate in the oral cavity (simian foamy viruses, *Retroviridae*; a cytomegalovirus and lymphocryptovirus; *Herpesviridae*; and alpha and gamma papillomaviruses, *Papillomaviridae*). None of the viruses identified have been shown to cause disease in chimpanzees or, to our knowledge, in humans. These data suggest that the risk of zoonotic viral disease from chimpanzee oral fluids in sanctuaries may be lower than commonly assumed.

## Introduction

Sanctuaries throughout Africa house and care for thousands of wild-born non-human primates (NHPs) that have been rescued from the illegal bushmeat and pet trades [1, 2]. In this setting, human caretakers typically have frequent, direct contact with NHPs with different geographic origins and histories which provides great opportunity for zoonotic pathogen transmission [3, 4]. One risk in such settings is NHP bites [5, 6]. NHP bites can cause severe physical injury, can become infected, and can transmit infectious agents, some of which are severely pathogenic in people [5, 7–11]. For example, herpes B virus, which infects macaques (*Macaca* spp.) often asymptomatically, is frequently lethal in humans [5]. Human infections

PRJNA888083 and BioSamples SAMN31207558 -
SAMN31207565. All other relevant data are within
the paper and supporting information files.

**Funding:** "This research was supported by National
Institutes of Health awards R37AG049395 and
R01AG049395 through the National Institute for
Aging (https://nia.nih.gov) and the Office of
Research on Women's Health (https://orwh.od.nih.
gov) to MET, AGR, and TLG and a University of
Wisconsin-Madison Global Health Institute (https://
ghi.wisc.edu) Graduate Student Research Award to
ED for research materials. ED was supported in the
form of salary by the University of Wisconsin-
Madison Comparative Biomedical Sciences
Training Grant T32OD010423 from the National
Institutes of Health Office of the Director (https://
nih.gov). The funders had no role in study design,
data collection and analysis, decision to publish, or
preparation of the manuscript."

**Competing interests:** The authors have declared
that no competing interests exist.

with apathogenic simian foamy viruses (SFVs) and with oncogenic primate T-lymphotropic viruses have resulted from NHP bites in hunters and occupational workers (e.g., people employed in research and zoo settings) [12–15]. Some viruses, such as Epstein-Barr virus of humans, are thought to be transmitted in saliva without an inciting bite [16, 17]. Furthermore, many viruses that can be detected in saliva can be transmitted through other routes (e.g., SARS-CoV-2: respiratory droplets [18]; herpes B virus: urine, feces, scratches [19]).

In addition to concern for virus transmission between sanctuary NHPs and sanctuary workers, the possible introduction and spread of viruses among and between NHP social groups and species warrants screening for pathogens that could impact relocation or reintroduction efforts [5, 20–23]. For example, human herpes simplex virus type 1, which has yet to be documented in wild mountain gorilla (*Gorilla beringei beringei*) populations, was detected in oral lesions of a sanctuary-housed juvenile eastern lowland gorilla (*G. b. graueri*) in Democratic Republic of Congo [20, 23].

The most commonly housed NHP at Pan African Sanctuary Alliance member sanctuaries is the chimpanzee (*Pan troglodytes*) [24, 25]. Thus, the majority of prior studies regarding sanctuary NHP health have been focused on chimpanzees [27]. Historically, research addressing viral infections in sanctuary-housed chimpanzees has been limited to using fecal and blood samples to determine what viruses are present in these populations [26, 27]. Thus far, these studies have shown that sanctuary chimpanzees are infected with a number of viruses that have rarely been associated with disease [28–30].

Despite the recognized risk of pathogen transmission, studies focused on viral infections in saliva from captive chimpanzees are few. Experimental infection of chimpanzees in research settings has suggested that saliva is a transmitting agent for certain human hepatitis viruses [31, 32]. In zoo- and laboratory-housed chimpanzees, there are reports of focal epithelial hyperplasia [33–35], a benign oral condition that has been associated with a papillomavirus, Pan paniscus papillomavirus type 1, first identified in zoo-housed bonobos (*Pan paniscus*) [36–38]. In wild chimpanzee populations, non-invasive saliva sampling has been recognized as a potentially valuable tool for pathogen detection, however, to our knowledge, only once has it been used to screen for a certain virus of interest (i.e., monkeypox virus) [39, 40].

When possible, the use of diverse and alternative sample types, such as saliva, may contribute to great ape health assessments by improving pathogen detection capabilities [16, 39–41]. There are limitations innate to all types of clinical samples regarding their suitability for the detection of pathogens due to differences in microbial properties, cellular and tissue tropism, and transmission routes [16, 42–44]. For example, rhinovirus C, a human respiratory pathogen that was associated with a lethal respiratory outbreak in wild chimpanzees in Uganda in 2013, is unlikely to be reliably detected in fecal samples due to virion characteristics that are not compatible with survival in the gastrointestinal tract [45]. Furthermore, certain enteric viruses (i.e., viruses of the gastrointestinal tract), which are known to be transmitted by the fecal-oral route, have been shown to replicate in the salivary glands and transmit through saliva in a mouse model [46]. Yet the results of viral diagnostics using saliva are likely to overlap with some of the more commonly utilized sample types. For example, certain viruses that are known to replicate in the oral cavity (e.g., SFV) can also be detected in peripheral blood mononuclear cells (an approach that has been employed in the chimpanzee sanctuary setting [6, 47]) and feces [48].

In this study, using metagenomic methods, we characterized viruses in saliva collected from two African sanctuary chimpanzee populations: A) Tchimpounga Chimpanzee Rehabilitation Centre (TCRC) in Republic of Congo and B) Ngamba Island Chimpanzee Sanctuary (NICS) in Uganda. Previously, we analyzed plasma samples from these same populations and found no evidence of pathogenic viruses and that sanctuary chimpanzees are infected with

many of the same types of viruses as their wild counterparts [26]. To our knowledge, this study is the first to identify viruses in saliva samples from sanctuary-housed chimpanzees which addresses the possibility that pathogenic viruses are being shed in saliva and could be transmitted to sanctuary personnel. Together with our prior surveillance using plasma, results from the current study could have implications for occupational health and safety of NHP sanctuary personnel, sanctuary management, and wild chimpanzee conservation efforts.

## Methods

### Ethics statement

Research at the sanctuaries was approved by the Institutional Animal Care and Use Committees at the University of Michigan (#8102) and Harvard University (#14-07-206-1). Additionally, research was approved by the Republic of Congo Ministry of Scientific Research and Technological Innovation and Jane Goodall Institute Congo for TCRC and the Uganda Wildlife Authority, the Uganda National Council for Science and Technology, and Chimpanzee Sanctuary and Wildlife Conservation Trust for NICS. Samples were shipped to the USA under Convention on International Trade in Endangered Species of Wild Fauna and Flora permits: Republic of Congo permit CG1126038 and USA permit 20US56953D/9 (TCRC); Uganda permit 004877 and USA permit 20US09881D/9 (NICS). Research complied with the standards outlined by the Pan African Sanctuary Alliance and adhered to the American Society of Primatologists Principles for the Ethical Treatment of Non-Human Primates.

### Study sites, study populations, and sample collection

The study sites were two Pan African Sanctuary Alliance member chimpanzee sanctuaries: A) TCRC and B) NICS (Fig 1). TCRC is located approximately 30km north of Pointe-Noire, Republic of Congo, in the Tchimpounga Nature Reserve, and cares for approximately 150 chimpanzees. NICS is located on Ngamba Island in Lake Victoria, Uganda, and cares for approximately 52 chimpanzees. After rescue, arrival at the sanctuaries, and rehabilitation, chimpanzees are integrated into social groups that semi-free range in forested enclosures during the day and stay in enclosed dens at night. At both sites, chimpanzees are provisioned with a variety of species-appropriate fruits, vegetables, and other foods multiple times per day, but the sanctuary chimpanzees are also able to forage within their forest enclosures.

At TCRC, we analyzed saliva from 1 sanctuary-born and 22 wild-born chimpanzees (11 females and 12 males, ages 7–31 years old) that was collected between July 4th and July 24th, 2019. At NICS, we analyzed saliva from 24 wild-born chimpanzees (13 females and 11 males, ages 7–30 years old) that was collected between July 29th and August 14th, 2016. Saliva samples were collected by researchers from chimpanzees who voluntarily allowed their mouth to be swabbed as in previous studies, following the same protocols [49–51]. To do so, the researchers first thoroughly washed their hands and then poured ground SweeTARTS (Ferrera Candy Company, Chicago, IL, USA) powder onto a cotton round. While holding the chimpanzee's bottom lip through the wire mesh enclosure, the cotton round was placed between the bottom lip and gums to absorb saliva. To minimize sample contamination, saliva was not collected from chimpanzees during provisioning periods (e.g., in the morning before breakfast at NICS, and either before breakfast or at least a half hour after feeding at TCRC), from chimpanzees who were observed eating anything remaining from prior feeding periods, or from chimpanzees who had visible cuts or other potential contamination sources in their mouths. After becoming saturated with saliva, a process that typically took 2–3 minutes and no more than 5 minutes, the cotton round was placed into a 10-mL syringe and the plunger was used to expel the saliva into a 1.2 mL cryogenic vial (Fisher Scientific, Waltham, MA, USA). TCRC saliva

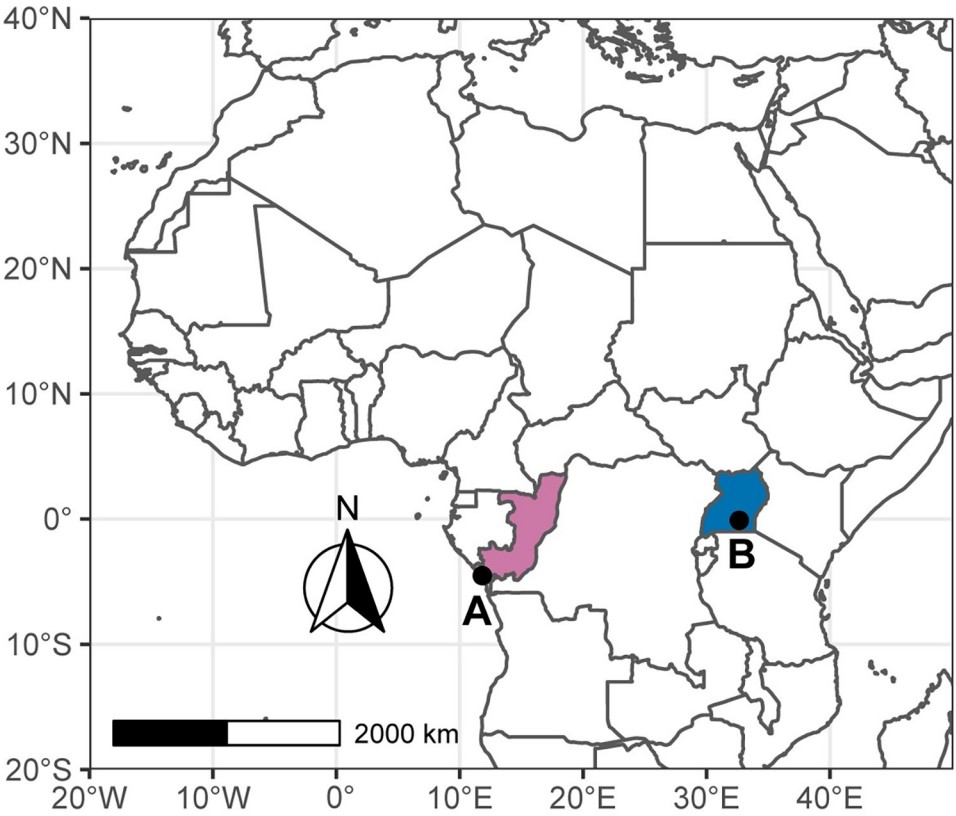

**Fig 1. Map of study sites.** (A) Tchimpounga Chimpanzee Rehabilitation Centre (TCRC) in Republic of Congo (purple) and (B) Ngamba Island Chimpanzee Sanctuary (NICS) in Uganda (blue). Map created using R v. 4.2.0 with Natural Earth (R Core Team, 2022).

samples were stored in the field in a cooler with ice packs and moved to a -20°C freezer within two hours. NICS saliva samples were frozen immediately in liquid nitrogen. All samples were kept frozen during shipment to the USA and were then stored at −80°C until processing.

## Sample preparation and sequencing

Following previously described methods optimized for virus detection in chimpanzee blood, feces, and respiratory swabs [4, 26, 45, 52–54], saliva samples were subject to metagenomic next-generation sequencing. First, we added 125ul of saliva to 125ul of Hank's Balanced Salt Solution (MilliporeSigma, Burlington, MA, USA), and the mixture was homogenized by bead beating and clarified by centrifugation. After nuclease digestion [55], we extracted nucleic acids using the QIAmp MinElute Virus Spin Kit (Qiagen, Hilden, Germany). We synthesized double-stranded cDNA using the SuperScript double-stranded cDNA Synthesis Kit (Invitrogen, Carlsbad, CA, USA). We purified the cDNA using AmpureXP beads (Beckman Coulter, Brea, CA, USA) and prepared DNA libraries using the Nextera XT DNA sample preparation kit (Illumina, San Diego, CA, USA). DNA libraries were sequenced on a MiSeq instrument (MiSeq Reagent Kit, V2 chemistry, 300 cycle kit; Illumina).

## Virus identification

We trimmed and prepared sequencing data using CLC Genomics Workbench v. 20.0.4 (Qiagen) as previously described [26]. First, we trimmed reads of low quality (Phred quality score <30)

and <50 bases in length. We then filtered out reads mapping to known contaminants and to a chimpanzee (*Pan troglodytes*) reference genome (GenBank accession number GCF_002880755.1). We performed de novo assembly of the remaining reads using SPAdes v. 3.15.2 with the metaSPAdes pipeline [56, 57]. We compared resulting contiguous sequences to the GenBank database using both the BLASTn (nucleotide) and BLASTx (protein) algorithms to identify viruses [58, 59]. We retained contigs representing viruses associated with mammalian hosts for further analyses (i.e., we excluded confirmed bacteriophage and viruses associated with invertebrates per ICTV reports). We used NOVOplasty to build out circular virus genomes [60].

To quantify viral abundances, we mapped reads from each individual to the sequence of a target gene (see Table 1 footnote) for each virus at a length fraction of 1.0 and similarity of 0.9 and classified an individual as positive for a virus if it had a normalized read count of $\geq 1$ read per million at a length of $\geq 50$ bases matching that virus. To estimate viral relative abundance, we determined the proportion of reads mapping to each virus and the proportion of reads mapping to any virus in the population (total viral abundance). We then normalized this measure to one million reads and to the length of the target sequence for each virus and applied a log transformation to calculate a metagenomic measure of viral abundance ($\log_{10}$ viral reads per million per kilobase of target sequence, or $\log_{10}$vRPM/kb) which is correlated with results from quantitative PCR assays [61] and has proven informative in our prior studies of chimpanzee viruses [26, 45, 52–54]. Complete papillomavirus genomes were annotated using PuMA [62].

## Phylogenetics and viral sequence comparisons

To infer phylogenetic relationships, we generated multiple sequence alignments of nucleotide sequences from the viruses identified in this study and known, related viruses in GenBank using the T-Coffee algorithm implemented by EMBL-EBI and hand edited the alignments as needed [63, 64]. We then inferred maximum-likelihood phylogenetic trees using PhyML with Smart Model Selection and 1000 bootstrap replicates [65, 66]. We displayed resulting phylogenetic trees in FigTree v. 1.4.4 [67].

We used MEGA X to calculate genetic distances (nucleotide p-distance ± standard error, 1000 bootstrap replicates) of viruses within and between the two sanctuary populations [68]. To compare the viruses identified in this study to each other and known relatives, we used Sequence Demarcation Tool v. 1.2 to generate pairwise alignments using the MUSCLE algorithm and calculate pairwise identities for nucleotide and amino acid sequences [69, 70].

## Statistical analyses

We conducted statistical analyses using R v. 4.2.0 implemented in R Studio [71, 72]. We calculated the prevalence of each virus (percentage of positive individuals) with modified Wald 95% confidence intervals (Agresti & Coull, 1998). We calculated odds ratios with 95% confidence intervals and performed two-tailed Fisher's exact tests ('fisher.test' in R) to assess the association between sex (male or female) and viral infection. We used Mann-Whitney U tests ('wilcox.test' in R) to assess the association between sex and viral abundance in infected individuals. We compared viral richness (number of viruses per individual) and total viral abundance between the two sanctuaries using Mann-Whitney U tests.

## Results

### Virus characterization

Next-generation sequencing produced an average of 1,533,717 reads per sample (SD ± 621,460) for TCRC and 1,589,629 (SD ± 203,431) for NICS after filtering for quality and

length. From these data, we identified 20 viruses of four genome types (ssRNA-RT, dsRNA, ssDNA, and dsDNA) (Table 1 and Fig 2). We identified 12 of these viruses at TCRC and 11 at NICS which are 66.5–99.6% similar to their closest match in GenBank based on BLASTn percent identity. All but one of these viruses, an unclassified CRESS (circular rep-encoding single-stranded) DNA virus identified at TCRC (TCPTV-13), were classified into five viral families: *Circoviridae*, *Herpesviridae*, *Papillomaviridae*, *Picobirnaviridae*, and *Retroviridae*. At least one virus from each of these families was identified at both sanctuaries. For the families *Herpesviridae* and *Papillomaviridae*, viruses from two genera (*Cytomegalovirus* and *Lymphocryptovirus*; *Alphapapillomavirus* and *Gammapapillomavirus*) were identified. Detailed pairwise sequence comparisons between viruses identified in this study and known relatives (S2–S9 Tables) as well as within- and between-population genetic distances (S10 Table) are provided in the supplementary information.

Simian foamy virus Pan troglodytes troglodytes (*Retroviridae*: *Simiispumavirus*; SFVptr) was identified at TCRC, and simian foamy virus Pan troglodytes schweinfurthii (SFVpsc) was identified at NICS (Fig 3). The TCRC SFVptr variant shares 96% Bet nucleotide sequence (NT) identity to known SFVptr variants (JQ867463, JQ867462, AF232917, AF232918) (S2 Table). The NICS SFVpsc variant shares 98% Bet NT identity with known SFVpsc variants (KX087159, U21247, EU381240).

We identified eight picobirnaviruses (*Picobirnaviridae*: *Orthopicobirnavirus*), all of which clustered phylogenetically in genogroup 1 except TCPTV-7 which clustered in genogroup 2 (S1 Fig). Picobirnavirus diversity within and between the two sanctuaries was similar (S3 and S10 Tables). Of the cycloviruses (*Circoviridae*: *Cyclovirus*) identified in this study (TCPTV-12, NAPTV-15, ChimpACyV2, NAPTV-17), two contain introns in their replicase genes (TCPTV-12 and ChimpACyV2), which has been reported for other cycloviruses [73] (S2 Fig and S4 Table). We identified the complete genome for ChimpACyV2 and NAPTV-17. ChimpACyV2 is most closely related to a cyclovirus identified in shrew feces (AB937982; [74]) to which it shares 75.4% genome-wide NT identity. Based on ICTV guidelines for cyclovirus species demarcation (<80% genome-wide NT identity; [73]), ChimpACyV2 represents a novel species. NAPTV-17 shares 91% genome-wide NT identity with a mongoose associated cyclovirus (MZ382573; [75]), a novel species that has not yet been formally classified [76], making NAPTV-17 a member of this putative species [73]. A single unclassified CRESS DNA virus (TCPTV-13) was identified at TCRC (S2 Fig and S2 and S5 Tables).

Pan paniscus papillomavirus type 1 (*Papillomavirus*: *Alphapapillomavirus*; PpPV1) was identified at each sanctuary (Fig 4A). The TCRC and NICS PpPV1 variants share 98.6% L1 NT identity (S6 Table). We also identified a novel alphapapillomavirus, Pan troglodytes papillomavirus type 1 (PtroPV1), at both sanctuaries. These variants share 99.9% L1 NT identity. PtroPV1 is most closely related to human papillomavirus type 177 (KR816168; 85% L1 NT identity) and represents a novel papillomavirus type based on the papillomavirus type demarcation threshold (<90% complete L1 NT identity) [77–79]. A single gammapapillomavirus (TCPTV-16; *Papillomaviridae*: *Gammapapillomavirus*) was identified at TCRC (Fig 4B and S7 Table).

We identified chimpanzee cytomegalovirus (CCMV) (*Herpesviridae*: *Cytomegalovirus*) at NICS which shares 95% DNA polymerase NT identity to the original CCMV (AF480884) and 81% to human cytomegalovirus (AY446894) (Fig 5A and S8 Table). Lastly, we identified Pan troglodytes lymphocryptovirus 1 (PtroLCV-1) (*Herpesviridae*: *Lymphocryptovirus*) at each sanctuary (Fig 5B). These variants share 99.8% DNA polymerase NT identity to each other, 99.0% to the original PtroLCV-1 (AF534226), and 92% to Epstein-Barr virus (AJ507799) (S9 Table).

**Table 1. Viruses in saliva from sanctuary chimpanzees at TCRC and NICS.**

| ID | Virus Name | Sanctuary | Abbreviation | Genome | Family[a] | Genus[a] | Closest match (source, location, year, accession)[b] | E-Value[b] | % ID (NT)[b] | Accession[c] |
|---|---|---|---|---|---|---|---|---|---|---|
| 1 | simian foamy virus Pan troglodytes schweinfurthii | NICS | SFVpsc | ssRNA-RT | *Retroviridae* | *Simiispumavirus* | Human foamy virus (human, Germany, EU381420) | 0 | 97.95 | OP934216 |
| 2 | simian foamy virus Pan troglodytes troglodytes | TCRC | SFVptr | ssRNA-RT | *Retroviridae* | *Simiispumavirus* | Human spumaretrovirus (chimpanzee, AF232918) | 0 | 96.53 | OP934220 |
| 3 | nabpantry virus 12 | NICS | NAPTV-12 | dsRNA (segmented) | *Picobirnaviridae* | *Orthopicobirnavirus* | Picobirnavirus sp. (human, Netherlands, 2016, OK562155) | 2.00E-87 | 66.54 | OP934217 |
| 4 | nabpantry virus 13 | NICS | NAPTV-13 | dsRNA (segmented) | *Picobirnaviridae* | *Orthopicobirnavirus* | Marmot picobirnavirus (marmot, China, 2013, KY928684) | 0 | 71.41 | OP934218 |
| 5 | nabpantry virus 14 | NICS | NAPTV-14 | dsRNA (segmented) | *Picobirnaviridae* | *Orthopicobirnavirus* | Porcine picobirnavirus (pig, USA, 2018, MW977424) | 0 | 75.72 | OP934219 |
| 6 | ticpantry virus 7 | TCRC | TCPTV-7 | dsRNA (segmented) | *Picobirnaviridae* | *Orthopicobirnavirus* | Chimpanzee picobirnavirus (chimpanzee, Sierra Leone, 2013–2016, MT350351) | 0 | 97.96 | OP934221 |
| 7 | ticpantry virus 8 | TCRC | TCPTV-8 | dsRNA (segmented) | *Picobirnaviridae* | *Orthopicobirnavirus* | Picobirnavirus sp. (cow, 2017–2019, China, MZ556513) | 0 | 73.46 | OP934222 |
| 8 | ticpantry virus 9 | TCRC | TCPTV-9 | dsRNA (segmented) | *Picobirnaviridae* | *Orthopicobirnavirus* | Porcine picobirnavirus (pig, USA, 2018, MW977506) | 0 | 87.26 | OP934223 |
| 9 | ticpantry virus 10 | TCRC | TCPTV-10 | dsRNA (segmented) | *Picobirnaviridae* | *Orthopicobirnavirus* | Picobirnavirus sp. (human, Cameroon, 2014, MH933806) | 0 | 81.72 | OP934224 |
| 10 | ticpantry virus 11 | TCRC | TCPTV-11 | dsRNA (segmented) | *Picobirnaviridae* | *Orthopicobirnavirus* | Human picobirnavirus (human, USA, 2018, OL875327) | 0 | 89.13 | OP934225 |
| 11 | nabpantry virus 15 | NICS | NAPTV-15 | ssDNA (circular) | *Circoviridae* | *Cyclovirus* | Cyclovirus Chimp53 (chimpanzee, Cameroon, 2003, GQ404881) | 0 | 95.15 | OP934203 |
| 12 | chimpanzee associated cyclovirus 2 | NICS | ChimpACyV2 | ssDNA (circular) | *Circoviridae* | *Cyclovirus* | Cyclovirus ZM36a (shrew, Zambia, 2012, AB937982) | 0 | 90.41 | OP934204 |
| 13 | nabpantry virus 17 | NICS | NAPTV-17 | ssDNA (circular) | *Circoviridae* | *Cyclovirus* | Mongoose-associated cyclovirus (mongoose, Saint Kitts and Nevis, 2017, MZ382573) | 0 | 93.17 | OP934205 |
| 14 | ticpantry virus 12 | TCRC | TCPTV-12 | ssDNA (circular) | *Circoviridae* | *Cyclovirus* | Swine cyclovirus (pig, Cameroon, 2012, KM392285) | 0 | 97.10 | OP934208 |

*(Continued)*

**Table 1.** (Continued)

| ID | Virus Name | Sanctuary | Abbreviation | Genome | Family[a] | Genus[a] | Closest match (source, location, year, accession)[b] | E-Value[b] | % ID (NT)[b] | Accession[c] |
|---|---|---|---|---|---|---|---|---|---|---|
| 15 | ticpantry virus 13 | TCRC | TCPTV-13 | ssDNA (circular) | Unclassified | Unclassified | Circovirus sp. (pig, China, 2017, MK377638) | 2.00E-92 | 70.64 | OP934209 |
| 16 | Pan paniscus papillomavirus type 1 | NICS | PpPV1 | dsDNA (circular) | *Papillomaviridae* | *Alphapapillomavirus* | Common chimpanzee papillomavirus 1 (chimpanzee, AF020905) | 0 | 95.94 | OP934206 |
| 17 | Pan troglodytes papillomavirus type 1 | NICS | PtroPV1 | dsDNA (circular) | *Papillomaviridae* | *Alphapapillomavirus* | Human papillomavirus type 177 (human, KR816168) | 0 | 84.16 | OP934207 |
| 18 | Pan paniscus papillomavirus type 1 | TCRC | PpPV1 | dsDNA (circular) | *Papillomaviridae* | *Alphapapillomavirus* | Common chimpanzee papillomavirus 1 (chimpanzee, AF020905) | 0 | 95.39 | OP934210 |
| 19 | Pan troglodytes papillomavirus type 1 | TCRC | PtroPV1 | dsDNA (circular) | *Papillomaviridae* | *Alphapapillomavirus* | Human papillomavirus type 177 (human, KR816168) | 0 | 84.16 | OP934211 |
| 20 | ticpantry virus 16 | TCRC | TCPTV-16 | dsDNA (circular) | *Papillomaviridae* | *Gammapapillomavirus* | Human papillomavirus (human, USA, 2015, MH777284) | 7.00E-152 | 78.55 | OP934212 |
| 21 | chimpanzee cytomegalovirus | NICS | CCMV | dsDNA (linear) | *Herpesviridae* | *Cytomegalovirus* | Panine betaherpesvirus 2 (chimpanzee, Germany, 2020, MZ151943) | 0 | 94.12 | OP934213 |
| 22 | Pan troglodytes lymphocryptovirus 1 | NICS | PtroLCV-1 | dsDNA (linear) | *Herpesviridae* | *Lymphocryptovirus* | Macaca arctoides gammaherpesvirus 1 (macaque, Georgia, 1985, MG471437) | 0 | 93.47 | OP934214 |
| 23 | Pan troglodytes lymphocryptovirus 1 | TCRC | PtroLCV-1 | dsDNA (linear) | *Herpesviridae* | *Lymphocryptovirus* | Pan paniscus lymphocryptovirus 1 (bonobo, AF534220) | 0 | 99.56 | OP934215 |

[a] Determined by phylogenetic analyses. See Figs 3–5 and S1, S2 Figs.

[b] Closest match, E-value, and percent identity (nucleotide) were identified by querying the target gene (Bet (*Retroviridae*), polymerase (*Picobirnaviridae*, *Herpesviridae*), Rep (*Circoviridae*), and L1 (*Papillomaviridae*)) nucleotide sequences against the NCBI's nonredundant nucleotide database using the discontiguous megablast homology searching algorithm. Length of target gene obtained for each virus and genome information is reported in S11 Table.

[c] GenBank accession number of viral sequence from this study.

## Viral prevalence

Overall, viral prevalence ranged from 4.2% to 87.5%. At TCRC, a cyclovirus and PtroPV1 were the most prevalent (53.3%; 95% confidence interval (CI): 33%, 70.8%). At NICS, PtroLCV-1 was the most prevalent virus (87.5% CI: 68.2%, 96.5%). The least prevalent viruses at TCRC were the gammapapillomavirus (TCPTV-16) and PpPV1, each infecting one individual (4.3% CI: <0.01%, 22.7%). At NICS, a cyclovirus (NAPTV-15) was the least prevalent (4.2% CI:

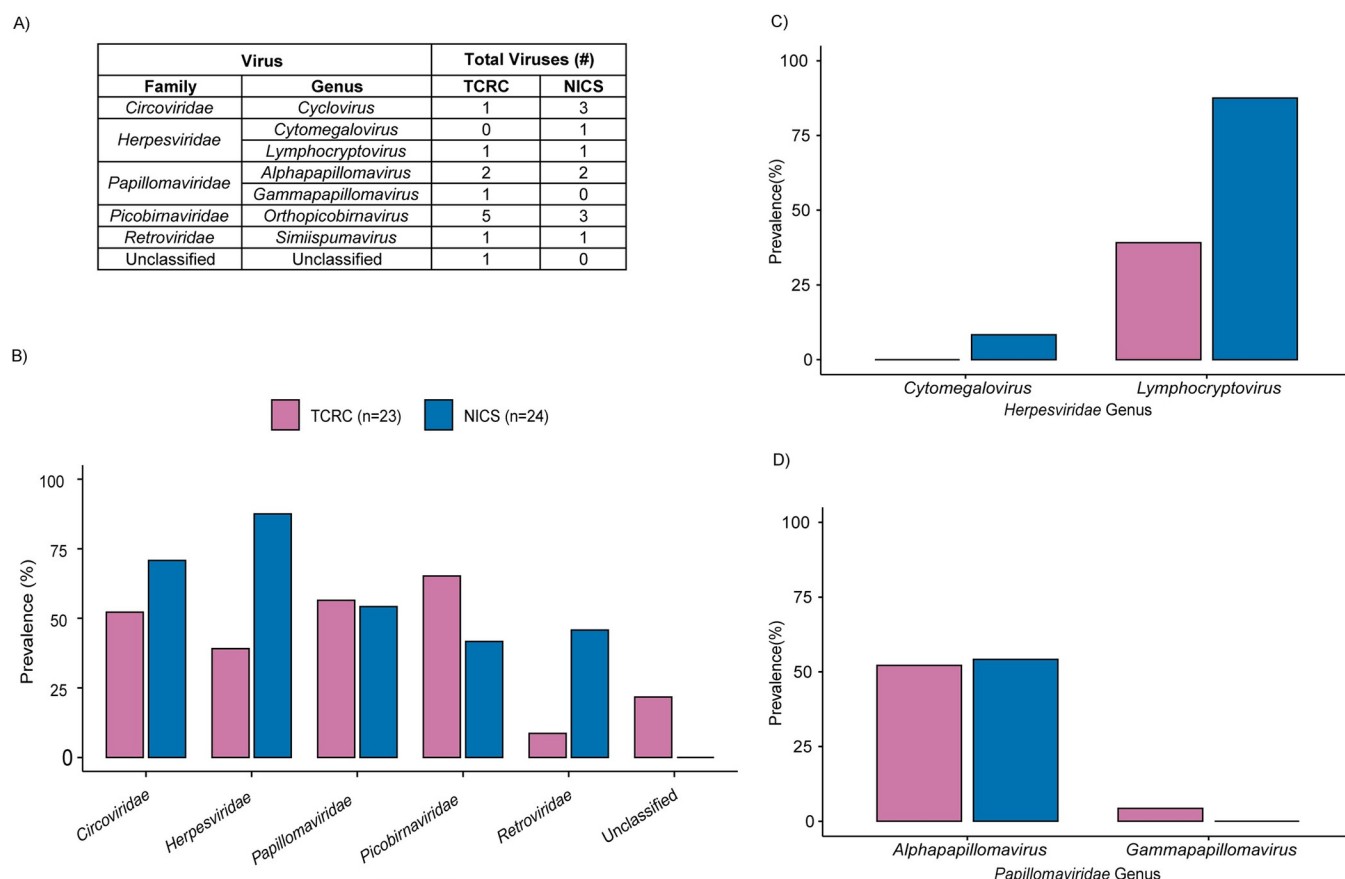

**Fig 2. Prevalence of viruses identified in saliva from chimpanzees at TCRC and NICS.** (A) Total number of viruses in each family and genus identified at TCRC and NICS. Family and genus were determined using phylogenetic analyses (Figs 3–5 and S1, S2 Figs). (B) Barplot displays the proportion (%) of individuals at each sanctuary who were positive for at least one virus in the family. (C) Barplot displays the proportion of individuals at each sanctuary who were positive for at least one virus in each identified *Papillomaviridae* genus. D) Barplot displays the proportion of individuals at each sanctuary who were positive for a virus in each identified *Herpesviridae* genus.

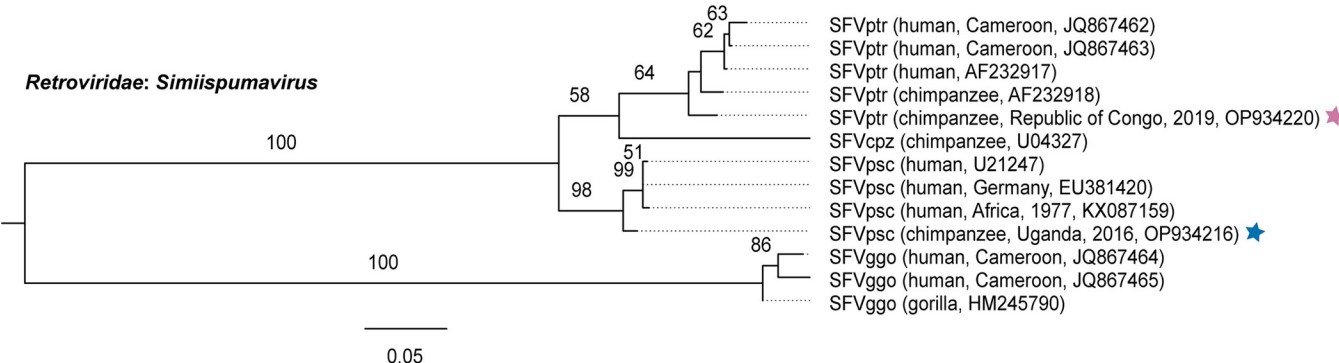

**Fig 3. Maximum-likelihood phylogenetic tree of simian foamy virus Bet gene nucleotide sequences.** Viruses identified in this study are marked with a colored star to indicate the sanctuary of origin (purple = TCRC, blue = NICS). Virus names are followed by (host, location, year, GenBank accession number). Bootstrap values ≥ 50% are represented by numbers beside branches (1000 replicates). Scale bar is equal to nucleotide substitutions per site.

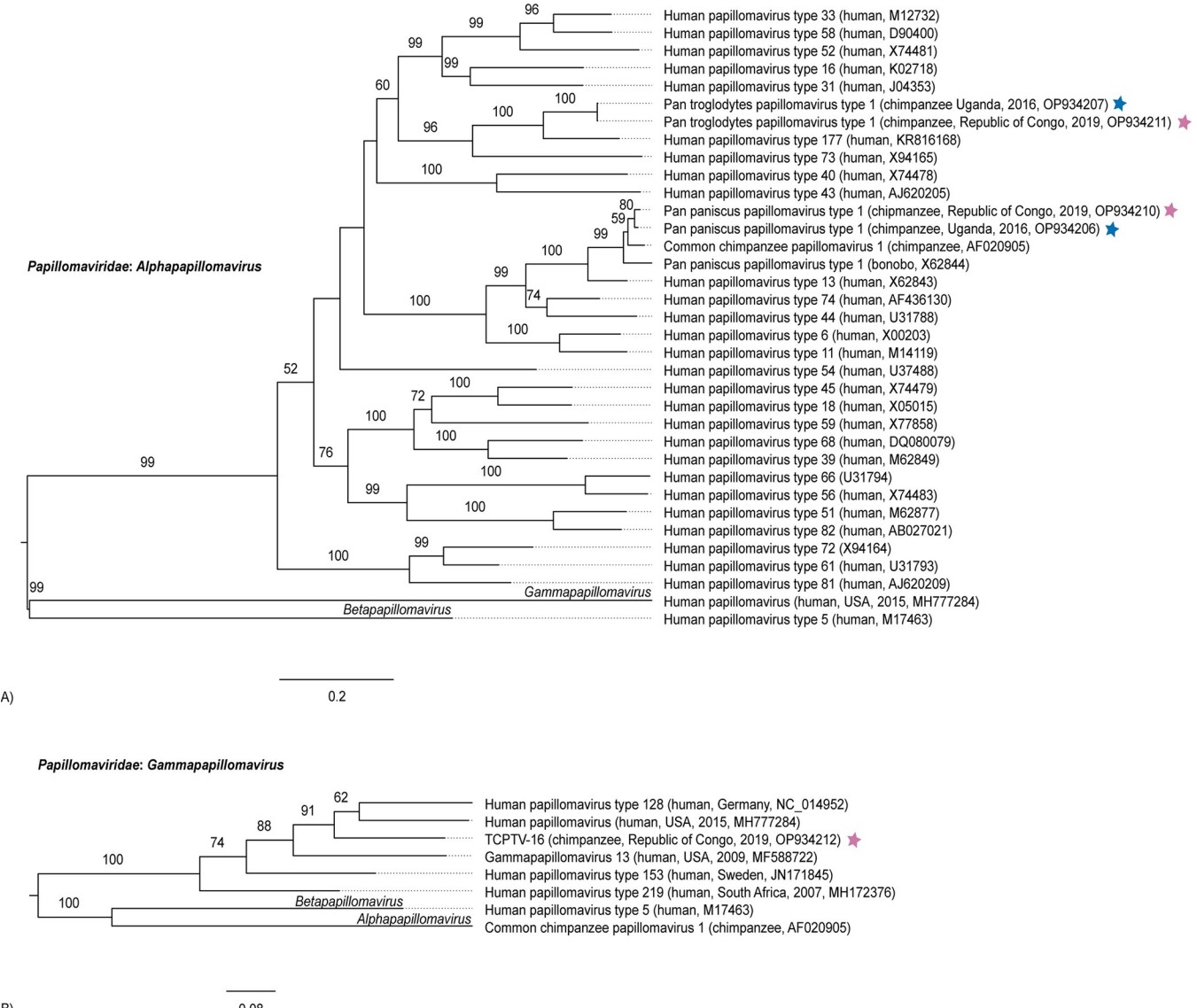

**Fig 4. Maximum likelihood phylogenetic trees of papillomavirus L1 gene nucleotide sequences.** (A) alphapapillomaviruses and (B) gammapapillomaviruses. Viruses identified in this study are marked with a colored star to indicate the sanctuary of origin (purple = TCRC, blue = NICS). Virus names are followed by (host, location, year, GenBank accession number). Bootstrap values ≥ 50% are represented by numbers beside branches (1000 replicates). Scale bar is equal to nucleotide substitutions per site.

<0.01%, 21.9%). Viral prevalence by family and genus is shown in Fig 2. The prevalence of individual viruses (overall and by sex) is listed in S12 Table. There were no statistically significant associations between viral presence and sex.

## Viral richness and abundance

Viral richness ranged from 0 to 7 at both sanctuaries (Fig 6A and S13, S14 Tables). The average viral richness was 2.8 (SD ± 1.8) at TCRC and 3.5 (SD ± 1.5) at NICS. Viral abundance ($\log_{10}$vRPM/kb) ranged among viruses and individuals from 0 to 4.22 at TCRC and 0 to 3.51 at NICS (Fig 6B, S3 Fig, and S13, S14 Tables). Among infected individuals, the virus with the highest abundance was NAPTV-15 (1.65), and the virus with the lowest abundance was

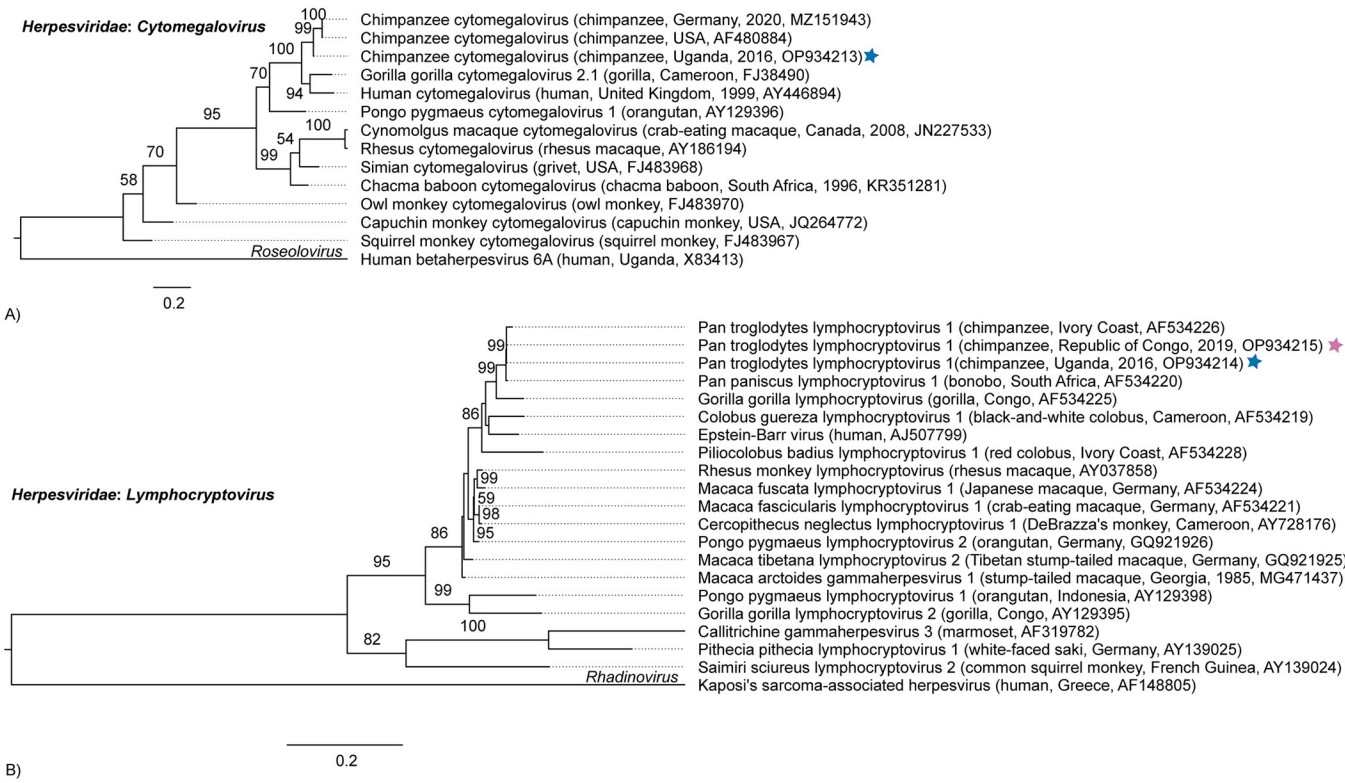

**Fig 5. Maximum likelihood phylogenetic trees of herpesvirus DNA polymerase catalytic subunit gene nucleotide sequences.** (A) cytomegaloviruses and (B) lymphocryptoviruses. Viruses identified in this study are marked with a colored star to indicate the sanctuary of origin (purple = TCRC, blue = NICS). Virus names are followed by (host, location, year, GenBank accession number). Bootstrap values ≥ 50% are represented by numbers beside branches (1000 replicates). Scale bar is equal to nucleotide substitutions per site.

NAPTV-17 (0.62), both cycloviruses. Total viral abundance (all viruses combined) was on average 0.63 (SD ± 0.7) at TCRC and 0.97 (SD ± 0.8), reaching a maximum of 3.23 at TCRC and 2.8 at NICS. We compared the distribution viral richness and total viral abundance at TCRC and NICS and did not observe a statistically significant difference (p-values: viral richness 0.19 and total viral abundance 0.07). For each sample, the number of sequencing reads mapping to each virus is reported in S15, S16 Tables.

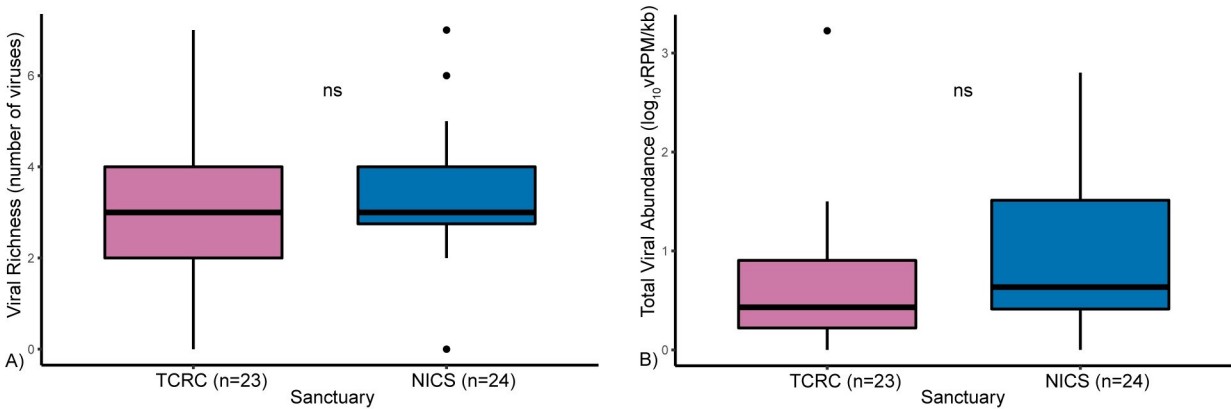

**Fig 6.** Boxplots of (A) viral richness and (B) total viral abundance for TCRC and NICS chimpanzees. Mann-Whitney U test comparisons between TCRC and NICS populations were not statistically significant (ns).

## Discussion

We identified a total of 20 viruses in saliva samples from two populations of wild-born, sanctuary-housed chimpanzees in Africa. Of the viruses identified, none are known to cause disease in chimpanzees or humans. Chimpanzees at these two sanctuaries were infected with viruses from the same families and genera including highly similar variants of two papillomaviruses and a herpesvirus (i.e., PpPV1, PtroPV1, and PtroLCV-1). With the exception of papillomaviruses, all types of viruses identified in this study have been documented previously in fecal samples from wild chimpanzees [48, 52, 53, 80–82]. Additionally, herpesviruses and SFV have been detected in tissues from deceased wild chimpanzees [83–87].

NHPs of many species are persistently infected with SFVs, which replicate in the superficial epithelial cells of the oral mucosa but do not cause known pathology [44, 88, 89]. Herpesviruses, also ubiquitous, naturally establish persistent, latent infections in primates and are intermittently shed in saliva [90, 91]. Several human herpesviruses (e.g., Epstein-Barr virus, human cytomegalovirus) are opportunistic pathogens, particularly in immunocompromised individuals [92, 93]. Mountain gorilla lymphocryptovirus has been associated with Epstein-Barr virus-like (human lymphocryptovirus) pathology in infant mountain gorillas [91]. Epstein-Barr virus-like pathology has not been documented for chimpanzees infected with PtroLCV-1. The true hosts of CRESS DNA viruses, including cycloviruses [73, 75, 76, 94], and picobirnaviruses [95, 96] are unknown, although arthropods and bacteria or fungi, respectively, are suspected. Thus, these viruses, neither of which are considered pathogenic, may actually represent dietary or environmental contaminants and not viral infection in chimpanzees [76, 95, 97, 98].

Papillomavirus infections can be persistent or transient and are usually asymptomatic or cause mild disease. Papillomaviruses have been detected in healthy skin swabs from zoo-housed chimpanzees and in oral cavity samples from laboratory-housed chimpanzees exhibiting focal epithelial hyperplasia [35, 99]. Previously, vaginal swabs from chimpanzees at NICS were tested for papillomaviruses using PCR, which yielded negative results [100]. Some papillomavirus types do cause disease (e.g., cervical cancer in humans and macaques) [101–103]. However, PpPV1 and PtroPV1 are distantly related to the carcinogenic human papillomavirus types, 16 and 18 (K02718 and X05015; 69–75% L1 NT identity; S6 Table) [104, 105]. In fact, PpPV1 clusters phylogenetically with members of the species *Alphapapillomavirus 10* which are associated with benign oral conditions such as focal epithelial hyperplasia in humans, chimpanzees, and bonobos [34–36, 77]. PtroPV1, the novel alphapapillomavirus identified in this study, is most closely related to human papillomaviruses of species *Alphapapillomavirus 11* including human papillomavirus type 73 which is classified as possibly carcinogenic to humans [105, 106].

Zoonotic transmission of SFVs has been documented both in the wild and in captivity [11, 14, 15, 47, 107, 108], but infections have not been shown to be pathogenic in the human host [88]. The SFVs detected in this study cluster phylogenetically with known SFV variants from central chimpanzees (*P. t. troglodytes*, SFVptr) and eastern chimpanzees (*P. t. schweinfurthii*, SFVpsc), consistent with virus-host co-speciation [109]. Herpesviruses are generally considered host-specific; however, the ability for cross-species transmission of herpesviruses between humans and NHPs is known for members of the subfamily *Alphaherpesvirinae* (i.e., herpes B virus, herpes simplex virus) [20, 110]. In contrast, transmission of cytomegaloviruses and lymphocryptoviruses between primates of different species (including humans) has not been documented, even where exposure would be high, such as in a chimpanzee and colobus monkey predator-prey system in the wild [81, 110–113]. Like most herpesviruses, papillomaviruses also demonstrate host-specificity [79]. Zoonotic transmission of papillomaviruses between chimpanzees and humans has not been reported.

Overall, our findings show that the sanctuary chimpanzees we analyzed are shedding viruses in their saliva that are known to replicate in the oral cavity but, to our knowledge, do not pose a risk to chimpanzee or human health. This conclusion is important because frequent contact between sanctuary apes and human staff has been an ongoing concern for sanctuary management [6]. Surveys of NHP workers in field, laboratory, zoo, or sanctuary settings have reported NHP bites in ~40% of workers [114, 115]. Contact with chimpanzee oral fluids could also occur during feeding, cleaning of enclosures, and especially during intensive rehabilitation of newly rescued individuals. Fortunately, our results suggest that the likelihood of sanctuary personnel acquiring pathogenic viral infections from chimpanzee saliva is low, at least in the sanctuaries we studied. Nevertheless, measures to minimize this risk should be taken particularly when viral infection status of sanctuary chimpanzees is unknown, which is often the case, and to avoid other sequelae (e.g., bacterial infection from bites [10]).

We emphasize that our results are specific to viruses in saliva, which may not reflect viruses in other body compartments. For example, we previously described viruses in plasma from TCRC and NICS chimpanzees (from primarily different individuals at TCRC but the same individuals at NICS) [27], but no viruses identified in this previous study of plasma were found in the present study of saliva (distinct picobirnaviruses were identified in both sample types) (S17 Table). We did, however, detect PtroLCV-1 at NICS, which has been previously detected in the blood buffy coat of these chimpanzees, but we did not detect herpesviruses in the genus *Rhadinovirus* which have also been previously detected at NICS [116]. Given that we detected other herpesviruses in this study, we suspect that the latter difference is due to intermittent shedding of herpesviruses in saliva [117, 118]. Likewise, SFV has also been detected in chimpanzees at NICS using buffy coat from blood [6]. Our results share little overlap with prior work using fecal samples from sanctuary chimpanzees to detect viruses which have primarily utilized virus-specific diagnostics (i.e., PCR). Notably, PtroLCV-1 was detected at lower prevalence in feces (3/21; 14.3%) [119] from NICS chimpanzees than in buffy coat (14/40; 35%) [116] or here in saliva (21/24; 87.5%). Lastly, we recognize that we are unable to comment on the infectivity and transmissibility of the viruses identified in this study as our methods do not distinguish between viral nucleic acids and infectious viruses [61].

Non-invasive saliva sampling has been utilized, albeit infrequently in comparison to fecal samples, to assess the health of wild chimpanzee populations. In wild western chimpanzees (*P. t. verus*), saliva collected non-invasively from food wadges has been used to detect *Staphylococcus* and *Streptococcus* spp. and monkeypox virus [40, 120–122]. Chewed ropes and vegetation have been used to detect herpesviruses and SFVs in saliva from wild mountain gorillas, olive baboons (*Papio anubis*), rhesus macaques (*M. mulatta*), and golden monkeys (*Cercopithecus kandti*) [23, 41, 91, 123]. Flavored chew swabs have been utilized to detect *Mycobacterium tuberculosis* in free-ranging macaques [124]. Our results suggest that metagenomic methods might be beneficial if combined with these or similar collection methods.

NHP sanctuary managers are faced with the need to develop long-term solutions to balance ongoing demand for care and capacity limits [125, 126]. In light of this, our findings add to prior studies that have shown that wild-born chimpanzees housed at sanctuaries appear broadly healthy with respect to viral infections [27], cardiovascular [127], endocrine [49, 51], and psychological health [51], in some ways similar to wild populations. We envision that our methods could be used to screen newly rescued individuals for viral pathogens upon arrival at a sanctuary, to prevent the introduction of orally transmitted pathogens, and to make decisions about husbandry and management of infected individuals, including decisions about relocation and reintroduction [24, 128].

## Supporting information

**S1 Fig. Maximum-likelihood phylogenetic tree of picobirnavirus RNA dependent RNA polymerase gene nucleotide sequences.** Viruses identified in this study are marked with a colored star to indicate the sanctuary of origin (purple = TCRC, blue = NICS). Virus names are followed by (host, location, year, GenBank accession number). Bootstrap values (%) $\geq$ 50 are represented by numbers beside branches (1000 replicates). Scale bar is equal to nucleotide substitutions per site.
(TIF)

**S2 Fig. Maximum-likelihood phylogenetic trees of cyclovirus and unclassified CRESS DNA virus replicase gene nucleotide sequences.** Viruses identified in this study are marked with a colored star to indicate the sanctuary of origin (purple = TCRC, blue = NICS). Virus names are followed by (host, location, year, GenBank accession number). Bootstrap values (%) $\geq$ 50 are represented by numbers beside branches (1000 replicates). Scale bar is equal to nucleotide substitutions per site.
(TIF)

**S3 Fig. Heatmap of saliva viral abundance of sanctuary chimpanzees at TCRC and NICS.** Displays viral abundance data ($\log_{10}$vRPM/kb) for each genus and total viral abundance data ($\log_{10}$vRPM/kb for all viruses) for each individual at each sanctuary. Values range from 0 (lightest) to 4.22 (darkest). [a] Genus refers to Table 1 and Fig 2. [b] For individuals infected with more than one virus from a genus, the average viral abundance is shown.
(TIF)

**S1 Table. Detailed sample inventory.**
(XLSX)

**S2 Table. Pairwise sequence comparisons of simian foamy viruses.**
(XLSX)

**S3 Table. Pairwise sequence comparisons of picobirnaviruses.**
(XLSX)

**S4 Table. Pairwise sequence comparisons of cycloviruses.**
(XLSX)

**S5 Table. Pairwise sequence comparisons of CRESS DNA viruses.**
(XLSX)

**S6 Table. Pairwise sequence comparisons of alphapapillomaviruses.**
(XLSX)

**S7 Table. Pairwise sequence comparisons of gammapapillomaviruses.**
(XLSX)

**S8 Table. Pairwise sequence comparisons of cytomegaloviruses.**
(XLSX)

**S9 Table. Pairwise sequence comparisons of lymphocryptoviruses.**
(XLSX)

**S10 Table. Within- and between-population (sanctuary) genetic distances of viruses identified in this study.**
(XLSX)

**S11 Table. Sequence information for viruses identified in this study.**
(XLSX)

**S12 Table. Prevalence and univariate statistical associations between sex (male or female) and prevalence and abundance of viruses in saliva from chimpanzees at TCRC and NICS.**
(XLSX)

**S13 Table. Viral richness and viral abundance for TCRC chimpanzees.**
(XLSX)

**S14 Table. Viral richness and viral abundance for NICS chimpanzees.**
(XLSX)

**S15 Table. Sequencing reads mapping to viruses identified in saliva from TCRC chimpanzees.**
(XLSX)

**S16 Table. Sequencing reads mapping to viruses identified in saliva from NICS chimpanzees.**
(XLSX)

**S17 Table. Reported viral diversity by ante-mortem sample type for sanctuary chimpanzees in Africa.**
(XLSX)

**S1 File. Reference list for S17 Table.**
(DOCX)

## Acknowledgments

We are grateful to the Republic of Congo Ministry of Scientific Research and Technological Innovation and Jane Goodall Institute Congo for research approval at TCRC as well as the Uganda Wildlife Authority, the Uganda National Council for Science and Technology, and Chimpanzee Sanctuary and Wildlife Conservation Trust for research approval at NICS. We thank Sofia Fernandez-Navarro, Rosemary Bettle, Alex Tumukunde, Titus Mukungu, and the animal caretakers and staff at both sites for assistance with data collection, and Emily Otali and Audrey Salvy for assistance with research permissions. We also thank Koenraad Van Doorslaer for assistance with papillomavirus genome annotation and virus naming and Christopher Dunn for assistance in the lab.

## Author Contributions

**Conceptualization:** Emily Dunay, Melissa Emery Thompson, Alexandra G. Rosati, Tony L. Goldberg.

**Formal analysis:** Emily Dunay.

**Funding acquisition:** Emily Dunay, Melissa Emery Thompson, Alexandra G. Rosati, Tony L. Goldberg.

**Investigation:** Emily Dunay, Joshua Rukundo, Rebeca Atencia, Megan F. Cole, Averill Cantwell, Melissa Emery Thompson, Alexandra G. Rosati, Tony L. Goldberg.

**Methodology:** Emily Dunay, Joshua Rukundo, Rebeca Atencia, Melissa Emery Thompson, Alexandra G. Rosati, Tony L. Goldberg.

**Resources:** Joshua Rukundo, Rebeca Atencia, Melissa Emery Thompson, Alexandra G. Rosati, Tony L. Goldberg.

**Supervision:** Melissa Emery Thompson, Alexandra G. Rosati, Tony L. Goldberg.

**Writing – original draft:** Emily Dunay, Tony L. Goldberg.

**Writing – review & editing:** Emily Dunay, Joshua Rukundo, Rebeca Atencia, Megan F. Cole, Averill Cantwell, Melissa Emery Thompson, Alexandra G. Rosati, Tony L. Goldberg.

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
