## [Decision Letter · Decision Letter 0]

3 May 2023

PONE-D-23-08923Viruses in saliva from sanctuary chimpanzees (*Pan troglodytes*) in Republic of Congo and UgandaPLOS ONE

Dear Dr. Dunay,

Thank you for submitting your manuscript to PLOS ONE. After careful consideration, we feel that it has merit but does not fully meet PLOS ONE’s publication criteria as it currently stands. Therefore, we invite you to submit a revised version of the manuscript that addresses the points raised during the review process.

Thank you for submitting your study for publication in PLoS One. We appreciate your confidence in our journal. Your manuscript has undergone extensive review by experts, and while they agree that your study would have a significant impact on the field, they have raised some concerns that require your attention before publication.

We kindly invite you to revise your manuscript and provide a detailed letter that addresses the comments of each reviewer point-by-point. Alternatively, you may raise a rebuttal if you disagree with any of the comments.

Thank you for considering PLoS One for the publication of your research.

We look forward to receiving your revised manuscript.

Kind regards,

Engin Berber

Academic Editor

PLOS ONE

Journal Requirements:

2. Please include a complete copy of PLOS’ questionnaire on inclusivity in global research in your revised manuscript. Our policy for research in this area aims to improve transparency in the reporting of research performed outside of researchers’ own country or community. The policy applies to researchers who have travelled to a different country to conduct research, research with Indigenous populations or their lands, and research on cultural artefacts. The questionnaire can also be requested at the journal’s discretion for any other submissions, even if these conditions are not met.  

Please find more information on the policy and a link to download a blank copy of the questionnaire here: https://journals.plos.org/plosone/s/best-practices-in-research-reporting. 

Please upload a completed version of your questionnaire as Supporting Information when you resubmit your manuscript.

4. Please expand the acronym “NIH, UW” (as indicated in your financial disclosure) so that it states the name of your funders in full.

"This research was supported by National Institutes of Health awards R37AG049395 and R01AG049395 through the National Institute for Aging (https://nia.nih.gov) and the Office of Research on Women's Health (https://orwh.od.nih.gov) to MET, AGR, and TLG and a University of Wisconsin-Madison Global Health Institute (https://ghi.wisc.edu) Graduate Student Research Award to ED. ED was supported by the University of Wisconsin-Madison Comparative Biomedical Sciences Training Grant T32OD010423 from the National Institutes of Health Office of the Director (https://nih.gov). The funders had no role in study design, data collection and analysis, decision to publish, or preparation of the manuscript."

We note that one or more of the authors is affiliated with the funding organization, indicating the funder may have had some role in the design, data collection, analysis or preparation of your manuscript for publication; in other words, the funder played an indirect role through the participation of the co-authors. If the funding organization did not play a role in the study design, data collection and analysis, decision to publish, or preparation of the manuscript and only provided financial support in the form of authors' salaries and/or research materials, please do the following:

(1) Review your statements relating to the author contributions, and ensure you have specifically and accurately indicated the role(s) that these authors had in your study. These amendments should be made in the online form.

(2) Confirm in your cover letter that you agree with the following statement, and we will change the online submission form on your behalf: 

7. We note that Figure 1 in your submission contain map images which may be copyrighted. All PLOS content is published under the Creative Commons Attribution License (CC BY 4.0), which means that the manuscript, images, and Supporting Information files will be freely available online, and any third party is permitted to access, download, copy, distribute, and use these materials in any way, even commercially, with proper attribution. For these reasons, we cannot publish previously copyrighted maps or satellite images created using proprietary data, such as Google software (Google Maps, Street View, and Earth). For more information, see our copyright guidelines: http://journals.plos.org/plosone/s/licenses-and-copyright.

(1) You may seek permission from the original copyright holder of Figure 1 to publish the content specifically under the CC BY 4.0 license.  

Reviewers' comments:

Reviewer's Responses to Questions

**Comments to the Author**

1. Is the manuscript technically sound, and do the data support the conclusions?

Reviewer #1: Yes

Reviewer #2: Yes

Reviewer #3: Yes

2. Has the statistical analysis been performed appropriately and rigorously? 

Reviewer #1: Yes

Reviewer #2: Yes

Reviewer #3: I Don't Know

3. Have the authors made all data underlying the findings in their manuscript fully available?

Reviewer #1: Yes

Reviewer #2: Yes

Reviewer #3: Yes

4. Is the manuscript presented in an intelligible fashion and written in standard English?

Reviewer #1: Yes

Reviewer #2: Yes

Reviewer #3: Yes

5. Review Comments to the Author

Reviewer #1: The paper for Dunay and collegues reports on viruses that are detected in saliva from chimpanzees housed in two sanctuaries in Africa, one in the Republic of Congo and the other in Uganda.

In both sanctuaries saliva was collected from 22 and 24 chimpanzees and subsequently analyzed by next generation metagenomic sequencing to document viral diversity in saliva, especially for the presence of viral families that include viruses that can be pathogenic for humans and and be transmitted to humans, especially animal keepers, via bites.

The methods are correct and are standard for this kind of studies. Moreover this is one of the first studies reporting on saliva. The authors identified around 20 viruses from 5 viral families, but none of them is currently associated with pathogenicity in chimpanzees or humans.

I have a few points that need to be clarified:

The authors mention also results on plasma or blood samples rom the same chimpanzee communities, can they specify whether this is also from the same animals for whom saliva samples are studies in this report?

Table 1 seems to contain important information, but the table is truncated and many information is not displayed. This should be rectified!!

The authors should provide more details on the number of reads for the different viruses they reported and the different gene fragments that are obtained. For some viruses , they obtained whole genomes, and for others they did phylogenetic analysis on genes fragments, but how much of the different virus genomes were recovered in the different samples?

Can the authors also provide information on what percentage of the total reads are represented by the viruses reported here, what is percentage of other viruses that were excluded and what was the overall diversity of viruses in saliva.

This can be useful information on whether saliva has a specific virome or whether some viruses are also observed in fecal samples from apes, which are more frequently studied, because these samples are easier to collect.

Reviewer #2: The authors present a manuscript based on the presence of viruses in saliva from african chimpanzees.

This manuscript explain that viruses found in saliva do not cause disease in chimpanzee and human. The authors used phylogenetic analysis for viruses identification and characterisation.

1- The authors have detected viruses in saliva but did not tell about the proviral load for all these viruses. This information is important to best understand the infectiosity of each virus and to best evaluate the risk for transmission to other non human primates and to human.

2- Some bibliographic references number are missed or used in duplicate or triplicate. The authors are requested to use a good logiciel for formating bibliography at the end of the manuscript.

3- Line 267, page 13: What does the sentence means:" Accesssion...this study."

4- Page 16, line 344: What does 70.8% represents? a CI?

5- At the line 419, the authors explain that "primates" include also humans. Is the word "primates" used several time in the manuscript include also "humans" or only animal? If so, the authors are requested to write NHPS and not only "primates".

6- Page 20, line 428: "...has been..." is written twice

Reviewer #3: The present study looks to use metagenomic methods to identify viruses in saliva samples from chimpanzees in two wildlife sanctuaries. I believe the study is interesting, well presented and valuable for practitioners in the field that work in wildlife rehabilitation facilities.

The abstract is succinct, and identifies the uniqueness of the study, as well as clearly summarizes the results and greater implications.

The authors do a great job introducing the subject material and highlighting the importance of the study.

The figures and tables visualize and support the findings well. There is a great number of both as supplementary material.

The methods are well detailed and comprehensively written so that others could repeat the methodology.

The discussion includes limitations and relevant comparisons with previous research. It presents relevant findings, and I believe the conclusions are within the realm of the results.

Minor revision: Please have a look over the reference list as some numbers are missing, repeated or out of order (5 and 6 missing, 21 repeated, 125 comes between 46 and 50).

6. PLOS authors have the option to publish the peer review history of their article (what does this mean?). If published, this will include your full peer review and any attached files.

Reviewer #1: No

Reviewer #2: No

Reviewer #3: No

---

## [Author Response · Author response to Decision Letter 0]

22 May 2023

Response to Reviewers

**All line numbers reference refer to Revised Manuscript with Track Changes**

Reviewers' comments:

Reviewer's Responses to Questions

Comments to the Author

1. Is the manuscript technically sound, and do the data support the conclusions?

Reviewer #1: Yes

Reviewer #2: Yes

Reviewer #3: Yes

2. Has the statistical analysis been performed appropriately and rigorously?

Reviewer #1: Yes

Reviewer #2: Yes

Reviewer #3: I Don't Know

3. Have the authors made all data underlying the findings in their manuscript fully available?

Reviewer #1: Yes

Reviewer #2: Yes

Reviewer #3: Yes

4. Is the manuscript presented in an intelligible fashion and written in standard English?

Reviewer #1: Yes

Reviewer #2: Yes

Reviewer #3: Yes

5. Review Comments to the Author

Reviewer #1: The paper for Dunay and collegues reports on viruses that are detected in saliva from chimpanzees housed in two sanctuaries in Africa, one in the Republic of Congo and the other in Uganda.

In both sanctuaries saliva was collected from 22 and 24 chimpanzees and subsequently analyzed by next generation metagenomic sequencing to document viral diversity in saliva, especially for the presence of viral families that include viruses that can be pathogenic for humans and and be transmitted to humans, especially animal keepers, via bites.

The methods are correct and are standard for this kind of studies. Moreover this is one of the first studies reporting on saliva. The authors identified around 20 viruses from 5 viral families, but none of them is currently associated with pathogenicity in chimpanzees or humans.

I have a few points that need to be clarified:

1 - The authors mention also results on plasma or blood samples rom the same chimpanzee communities, can they specify whether this is also from the same animals for whom saliva samples are studies in this report?

Response to Comment 1: We are grateful to the reviewer for making this suggestion. In order to address the reviewer's critique, we have added information to the Discussion stating whether the same individuals were sampled in the previous plasma manuscript and the current saliva manuscript (lines 444-445).

2 - Table 1 seems to contain important information, but the table is truncated and many information is not displayed. This should be rectified!!

Response to Comment 2: We have followed the PLOS submission guidelines for Tables as listed on the website (https://journals.plos.org/plosone/s/tables), specifically, “Do not split your table or otherwise try to make the table appear within the manuscript margins if it does not fit on one page. In Word, tables that run off of the manuscript page can be seen using Draft View.” If you would like us to submit the table in a different format for better viewing, please advise what is best. 

3 - The authors should provide more details on the number of reads for the different viruses they reported and the different gene fragments that are obtained. For some viruses , they obtained whole genomes, and for others they did phylogenetic analysis on genes fragments, but how much of the different virus genomes were recovered in the different samples?

4 - Can the authors also provide information on what percentage of the total reads are represented by the viruses reported here, what is percentage of other viruses that were excluded and what was the overall diversity of viruses in saliva. This can be useful information on whether saliva has a specific virome or whether some viruses are also observed in fecal samples from apes, which are more frequently studied, because these samples are easier to collect.

Response to Comments 3 and 4: We are grateful to the reviewer for making these suggestions. This is indeed useful information. To address these points, we have created additional supporting information tables (S11, S15, and S16). S11 Table (now referenced in line 268) lists for each virus what gene was utilized for phylogenetic analyses and viral abundance calculations, its length, whether the complete or partial gene was obtained, and for which viruses we obtained the complete genome. Additionally, full sequence data (partial or complete genome) for each virus is annotated and available on GenBank (see corresponding accession numbers in Table 1). S15 Table (TCRC) and S16 Table (NICS), both now referenced in lines 366-367, provide the number of reads from each sample that mapped to each virus and what percentage of total reads after quality trim and removal of contaminants/host are viral for each sample. Lastly, as mentioned in the discussion section all types of viruses identified in this study with the exception of papillomaviruses have been documented in the feces of wild chimpanzees (lines 379-382) and one, PtroLCV-1, has been documented in feces from sanctuary chimpanzees at NICS (lines 454-457). This information is further detailed in the S17 Table. Reported viral diversity by ante-mortem sample type for sanctuary chimpanzees in Africa.

Reviewer #2: The authors present a manuscript based on the presence of viruses in saliva from african chimpanzees.

This manuscript explain that viruses found in saliva do not cause disease in chimpanzee and human. The authors used phylogenetic analysis for viruses identification and characterisation.

1- The authors have detected viruses in saliva but did not tell about the proviral load for all these viruses. This information is important to best understand the infectiosity of each virus and to best evaluate the risk for transmission to other non human primates and to human

Response to Comment 1: We thank the reviewer for raising this important point. We agree that it would be useful to understand the infectivity and risk of transmission of these viruses. We have therefore added a statement to the Discussion explaining this idea and how it applies to our study (lines 458-461), and we have provided supporting citations.

2- Some bibliographic references number are missed or used in duplicate or triplicate. The authors are requested to use a good logiciel for formating bibliography at the end of the manuscript.

Response to Comment 2: The reviewer is indeed correct. We have gone through the manuscript with a fine-tooth comb to ensure that all references are properly formatted.

3- Line 267, page 13: What does the sentence means:" Accesssion...this study."

Response to Comment 3: This is an excellent point. This sentence means that the accession number listed in Table 1 is the corresponding GenBank accession number corresponding to the nucleotide sequence of each virus identified in this study. We have changed the wording of this sentence to clarify this point (line 269).

4- Page 16, line 344: What does 70.8% represents? a CI?

Response to Comment: Yes, this represents the upper limit of the 95% confidence interval as defined and indicated throughout the manuscript: (53.3%; 95% confidence interval (CI): 33%, 70.8%).

5- At the line 419, the authors explain that "primates" include also humans. Is the word "primates" used several time in the manuscript include also "humans" or only animal? If so, the authors are requested to write NHPS and not only "primates".

Response to Comment 5: This is indeed a valid point, many thanks. Based on the reviewer's excellent suggestion, we have reviewed the entire manuscript and made changes to reflect appropriate use of “NHP” and “primate” throughout (lines 48 and 433).

6- Page 20, line 428: "...has been..." is written twice

Response to Comment 6: Good catch! We have fixed the typo by removing the duplication (line 432).

Reviewer #3: The present study looks to use metagenomic methods to identify viruses in saliva samples from chimpanzees in two wildlife sanctuaries. I believe the study is interesting, well presented and valuable for practitioners in the field that work in wildlife rehabilitation facilities.

The abstract is succinct, and identifies the uniqueness of the study, as well as clearly summarizes the results and greater implications.

The authors do a great job introducing the subject material and highlighting the importance of the study.

The figures and tables visualize and support the findings well. There is a great number of both as supplementary material.

The methods are well detailed and comprehensively written so that others could repeat the methodology.

The discussion includes limitations and relevant comparisons with previous research. It presents relevant findings, and I believe the conclusions are within the realm of the results.

Response: Many thanks for the positive comments!

Minor revision: Please have a look over the reference list as some numbers are missing, repeated or out of order (5 and 6 missing, 21 repeated, 125 comes between 46 and 50).

Response: We thank the reviewer for raising this important point. To address this point, we have carefully looked over the entire reference list and in-text citations and ensured that numbering is now correct throughout.

6. PLOS authors have the option to publish the peer review history of their article (what does this mean?). If published, this will include your full peer review and any attached files.

Do you want your identity to be public for this peer review? For information about this choice, including consent withdrawal, please see our Privacy Policy.

Reviewer #1: No

Reviewer #2: No

Reviewer #3: No

---

## [Decision Letter · Decision Letter 1]

19 Jun 2023

Viruses in saliva from sanctuary chimpanzees (*Pan troglodytes*) in Republic of Congo and Uganda

PONE-D-23-08923R1

Dear Dr. Dunay,

We’re pleased to inform you that your manuscript has been judged scientifically suitable for publication and will be formally accepted for publication once it meets all outstanding technical requirements.

Kind regards,

Engin Berber

Academic Editor

PLOS ONE

Additional Editor Comments (optional):

Reviewers' comments:

Reviewer's Responses to Questions

**Comments to the Author**

1. If the authors have adequately addressed your comments raised in a previous round of review and you feel that this manuscript is now acceptable for publication, you may indicate that here to bypass the “Comments to the Author” section, enter your conflict of interest statement in the “Confidential to Editor” section, and submit your "Accept" recommendation.

Reviewer #2: All comments have been addressed

Reviewer #3: All comments have been addressed

2. Is the manuscript technically sound, and do the data support the conclusions?

Reviewer #2: Yes

Reviewer #3: Yes

3. Has the statistical analysis been performed appropriately and rigorously? 

Reviewer #2: Yes

Reviewer #3: Yes

4. Have the authors made all data underlying the findings in their manuscript fully available?

Reviewer #2: Yes

Reviewer #3: Yes

5. Is the manuscript presented in an intelligible fashion and written in standard English?

Reviewer #2: Yes

Reviewer #3: Yes

6. Review Comments to the Author

Reviewer #2: (No Response)

Reviewer #3: (No Response)

7. PLOS authors have the option to publish the peer review history of their article (what does this mean?). If published, this will include your full peer review and any attached files.

Reviewer #2: No

Reviewer #3: No

---

## [Editor Report · Acceptance letter]

22 Jun 2023

PONE-D-23-08923R1 

Viruses in saliva from sanctuary chimpanzees (*Pan troglodytes*) in Republic of Congo and Uganda 

Dear Dr. Dunay:

I'm pleased to inform you that your manuscript has been deemed suitable for publication in PLOS ONE. Congratulations! Your manuscript is now with our production department. 

Kind regards, 

on behalf of

Dr. Engin Berber 

Academic Editor

PLOS ONE